# Screening of Precancerous Lesions in Women with Human Papillomavirus (HPV) Infection by Molecular Typing and MicroRNA Analysis

**DOI:** 10.3390/jpm13030531

**Published:** 2023-03-15

**Authors:** Serena Varesano, Alessandra Pulliero, Emanuele Martorana, Gabriele Pizzino, Gabriele Raciti, Simona Coco, Valerio Gaetano Vellone, Alberto Izzotti

**Affiliations:** 1IRCCS Ospedale Policlinico San Martino, 16132 Genoa, Italy; 2Department of Health Sciences, University of Genoa, 16132 Genoa, Italy; 3Istituto Oncologico del Mediterraneo, 95029 Viagrande, Italy; 4Dipartimento di Scienze Biomediche, Odontoiatriche e delle Immagini Morfologiche e Funzionali (BIOMORF), Università degli Studi di Messina (ME), 98122 Messina, Italy; 5Lung Cancer Unit, IRCCS Ospedale Policlinico San Martino, 16132 Genoa, Italy; 6Department of Integrated Surgical and Diagnostic Sciences (DISC), University of Genoa, 16132 Genoa, Italy; 7Fetal and Perinatal Pathology Unit, IRCCS Istituto Giannina Gaslini, Via Gerolamo Gaslini, 16147 Genoa, Italy; 8Department of Experimental Medicine, University of Genoa, 16132 Genoa, Italy

**Keywords:** microRNAs, human papilloma virus HPV, cancer prevention, personalized medicine

## Abstract

**Simple Summary:**

Cervical cancer is the second most common cancer among women aged 15 to 44, affecting more than 500,000 women each year. Therefore, is important to implement preventive measures, based on screening programs, which allow precancerous lesions to be identified and treated early before they evolve into cancer. The aim of the proposed study is to implement the routine diagnostics of HPV precancerous cervical lesions by introducing new molecular diagnostic tools. The microRNA analysis panel can improve early diagnosis, understand the nature of the lesion and, consequently, improve the clinical management of patients with HPV precancerous cervical lesions. Genotyping, the determination of viral load, the evaluation of viral integration status and the expression profile of microRNAs were examined for this purpose. A characterization of lesions before diagnostic interventions allows to treat in a targeted way only lesions with a high risk of cancer progression, improving early diagnosis. The results allow the stratification of the risk of progression, the creation of personalized, therapeutic and follow-up protocols.

**Abstract:**

Human papillomavirus (HPV) is causatively associated with cervical cancer, the fourth most common malignant disease of women worldwide: (1) The aim of the proposed study is to implement routine diagnostics of HPV precancerous cervical lesions by introducing new molecular diagnostic tools. (2) Methods: This is a retrospective cohort study with a total of twenty-two formalin-fixed paraffin-embedded (FFPE) cervical samples of various sample type (nine biopsy and thirteen conization) each patient had a previous abnormal results of pap test or HPV DNA test. Genotyping, viral load and co-infections were determined. For each patient, the individual expression of 2549 microRNAs were evaluated by microarray and qPCR. (3) Results: Our data demonstrates that the microRNAs were commonly expressed in tissues biopsies. miR 4485-5p, miR4485-3p and miR-4497 were highly down-regulated in tissue biopsies with HPV precancerous cervical lesions. (4) Conclusions: the introduction of a microRNA analysis panel can improve early diagnosis, understand the nature of the lesion and, consequently, improve the clinical management of patients with HPV precancerous cervical lesions.

## 1. Introduction

Human papillomavirus (HPV) is the primary cause of cervical cancer, the first cancer to date recognized by the World Health Organization (WHO) as attributable to an infection. Worldwide, WHO indicates cervical cancer as the fourth most frequent cancer in women with more of six hundred thousand new cases in 2021 [1]. Cervical cancer continues to be a relevant health problem. In Italy, it represents the fifth cancer by frequency in women under 50 years of age. In 2020, there were about 3500 new diagnoses of cervical cancer and over 1500 women die from this tumor [2]. An effective preventive strategy is to invest in screening programs that allow precancerous lesions to be identified and treated early before they turn into cancer.

International Agency for Research on Cancer [3], identify 12 genotypes as high-risk oncogenic (HPV 16, 18, 31, 33, 35, 39, 45, 51, 52, 56, 58 and 59) [4]. HPV16 and HPV18 are the types most frequently associated with cervical lesions. More than 50% of cervical neoplasms are caused by HPV 16 infection and about 20% by HPV 18. HPV 16 has been detected in both low-grade lesion and cervical cancer [5]. HPV is a virus with double-stranded circular DNA, characterized by genes that express proteins needed for DNA replication, transcription or for viral assembly and release. HPV genome contains the early region’s genes, which code for regulatory proteins (E1-E7) that represent the main oncoproteins in HPV, whose overexpression is a prerequisite for the development of HPV tumor [6]. Cervical cancer is characterized by a premalignant phase that can be detected by the cytological examination of exfoliated cervical cells and confirmed by the histological examination of cervical material. Premalignant changes are reflected in a spectrum of histological abnormalities ranging from cervical intraepithelial neoplasia grade 1 (CIN 1) or mild dysplasia to moderate dysplasia (CIN 2) and severe dysplasia or carcinoma in situ (CIN 3) [7]. CIN 1 is considered as a morphologic expression of HPV infection and CIN 2 as a mixture of CIN 1 and CIN 3, frequently regressing. High grade squamous intraepithelial lesion is a squamous cell abnormality associated with HPV. It includes the used terms of CIN 2, CIN 3, moderate and severe dysplasia, and carcinoma in situ. This current terminology for high grade squamous intraepithelial lesion was introduced by the Bethesda System for Reporting Cervical Cytology for cytology specimens in 1988, and has since been adopted for histology specimens by the Lower Anogenital Squamous Terminology Standardization Consensus Conference [8] and the World Health Organization (WHO) in 2012 and 2014, respectively. Though not all high grade squamous intraepithelial lesion will progress to cancer, it is considered a pre-cancerous lesion and therefore is usually treated aggressively. It is estimated that the risk of high-risk HPV infections is about 80%, but most infections are naturally eliminated by the host’s immune system [7]. Only lesions caused by the persistent infections of high-risk genotypes can develop into invasive cancer. However, high grade precancerous lesions (CIN 2–CIN 3) take a long time to develop into invasive cancer, sometimes 20–30 years [8]. The long period of time allows the implementation of preventive strategies for the identification and treatment of cervical lesions, preventing their cancerous progression. Unfortunately, although it is preventable, cervical cancer still affects many women [9].

WHO recommends to screening all women over 25–30 years old with pap test and HPV DNA test to identify precancerous lesions, which are usually asymptomatic, before they progress to invasive cancer [10]. To date, pap tests and HPV DNA tests are the benchmark tests of the first level, but the number of false negatives and false positives tests is not negligible [11,12]. However, with morphological examinations alone, it is not possible to determine the risk of progression of the lesions. Although oncogenic genotypes are associated with cervical cancer, HPV infection alone is not sufficient to cause a malignant transformation. It is necessary to consider multilevel molecular and epigenetic factors, such as microRNA expression profile, viral integration status, viral genotype and viral load. Recent studies show how the oncogenic or tumor suppressor role of microRNAs can influence cell differentiation, cell cycle regulation and apoptosis in carcinogenesis [13]; as well as regulate the post-transcriptional expression of some genes. The role of microRNAs has also been highlighted in cervical cancer, although, as is well known, its primary cause is the persistence of high-risk HPV infection [14]. Many studies indicate that microRNA deregulation contributes to cervical cancer tumorigenesis [15]. MicroRNA alterations drive the progression of cervical cancer from CIN 1 to full blown cancer [16]. The importance of microRNAs in cervical tumors is linked to the fact that the microRNA loci are associated with fragile sites, known as insertion sites of the HPV virus in cervical tumors. Furthermore, the genes encoded by the virus can influence the expression of microRNAs in cervical cells. In general, microRNAs can regulate both the tumor suppressor and oncogenes genes and the altered expression of microRNAs represents an early event in the induction of carcinogenic by HPV infection [17]. Many studies show that in cervical cancer the expression of some microRNAs increases (miR-20a, miR-20b, miR-93, miR-224) and decreases in others (miR-127, miR-143/145, miR-218) [9,18]. Other studies have shown that miR-218 was higher in high-risk HPV + than in high-risk HPV negative carcinomas; furthermore, miR-146a levels were lower in p16INKa-positive (marker of high-risk HPV infection) than in p16INKa-negative samples [19]. It has also been shown that high-risk HPV infection is associated with a significant reduction in the expression of miR34a/b/c, miR-218, miR-210 and let-7 family microRNAs [20]. Although oncogenic genotypes are associated with cervical cancer, HPV infection alone is not sufficient to induce malignant and cervical cancer transformation, which also requires the consideration of multilevel molecular factors, such as viral integration status, genotyping, viral load and microRNA expression profiles. The aim of the proposed study is to implement the routine diagnostics of HPV precancerous cervical lesions by introducing new molecular diagnostic tools. The introduction of a microRNA analysis panel can improve early diagnosis, understand the nature of the lesion and, consequently, improve the clinical management of patients with HPV precancerous cervical lesions. The molecular characterization of preneoplastic cervical lesions would allow to stratify the risk of progression towards invasive forms and consequently to personalize the follow-up and therapeutic interventions.

The comparison of microRNA expression between normal samples and CIN 1–3/invasive carcinoma samples has been already reviewed in our previous paper [16]. The specific and original goal of this study is to focus on the microRNA driving the progression from CIN 1 to CIN 2/3.

The aim of the current study is to evaluate the feasibility and utility of including epigenetic alterations associated with microRNA in molecular cervical cancer screenings in order to achieve personalized preventive programs.

## 2. Materials and Methods

### 2.1. Tissue and Serum Specimens

This is a retrospective cohort study with a total of 22 formalin-fixed paraffin-embedded (FFPE) cervical samples of various sample type (9 biopsy and 13 conization) diagnosed between years 2017–2021 at the Division of Histopathology and Cytopathology of Hospital Policlinico San Martino in Genova, Italy. Patients were aged between 26–63 years (median = 38 years) and each patient had previously received abnormal pap test and/or HPV DNA test results.

All patients were diagnosed through the examination of hematoxylin- and eosin-stained sections combined with the immunohistochemistry of p16 and ki67. The experimental molecular analysis of viral integration status, viral genotype and viral load was developed and performed at the Division of Hygiene, Hospital Policlinico San Martino in Genova, Italy; while the microRNA expression profile was developed and completed at Division of Mutagenesis and Cancer Prevention, Hospital Policlinico San Martino in Genova, Italy. Each participant provided written informed consent, which was approved by the Liguria Regional Ethics Committee (P.R. 162REG2017).

### 2.2. RNA Extraction from FFPE

Two to four 3 μm-thick sections were cut from FFPE tissue samples and deparaffinized using the deparaffinization solution (Qiagen, Hilden, Germany) followed by proteinase K digestion as protocol indication. microRNAs were extracted using microRNAeasy FFPE kit (Qiagen, Hilden, Germany), according to the manufacturer’s protocol. The extracted RNA was ultimately eluted in 30 μL of RNase free water. The concentration and purity of the isolated RNA was evaluated by Bioanalyzer (Agilent technology, Santa Clara, CA, USA).

### 2.3. Immunohistochemistry

The immunohistochemistry was carried out by an automatic immune-stainer Ventana Benchmarck XT (Ventana Medical System Inc., Innovation Park Dr, Oro Valley, AZ, USA), according to the established protocol. The antibodies used in this study were: anti-p16^INK4a^ (monoclonal, clone E6H4, Roche), pre-diluted and incubated at 37 °C for 20 min, and anti-human Ki-67 (monoclonal, Rabbit clone Anti-Human Ki-67 SP6, Roche), pre-diluted and incubated at 37 °C for 16 min. The reaction was developed using ultra-view Universal DAB Detection kit (Ventana Medical Systems Inc., USA) and counterstained with Gill’s Modified Hematoxylin (Ventana Medical Systems Inc., USA) for 8 min at room temperature, followed by 4 min of Bluing-reagent (Ventana Medical Systems Inc., USA).

### 2.4. Evaluation of Tumor Section

The evaluation of immune-stained tumor sections was performed by an experienced pathologist. For the evaluation of both immune staining, the dysplastic epithelium was divided into three thirds from the basal to the luminal side. For p16, only intense, diffuse, nuclear and cytoplasmic staining extending at least to the middle third “block positivity” was considered positive. For ki67, only the nuclear stain was considered.

### 2.5. DNA Extraction from FFPE and Determination of Genotyping, Viral Load and Co-Infections by the Multiplex Real-Time PCR

Two to four 3 μm-thick sections were cut from FFPE tissue samples and placed in Eppendorf Tube^®^ 1.5 mL. Subsequently, deparaffined using the MagCore^®^ Genomic DNA FFPE One-Step Kit (RBC Bioscience Corp. New Taipei City, Taiwan), following the manufacturer’s instructions (cartridge code: 405; execution time: 16 h; elution volume: 60 µL) using an extractor automated MagCore^®^ HF16 Plus nucleic acid (RBC Bioscience Corp.). Genotyping, viral load and co-infections were detected by Anyplex II HPV28 kit (Seegene, Seoul, Republic of Korea), which simultaneously identifies 28 genotypes: 19 high-risk HPV types (16, 18, 26, 31, 33, 35, 39, 45, 51, 52, 53, 56, 58, 59, 66, 68, 69, 73 and 82) and 9 low-risk HPV types (6, 11, 40, 42, 43, 44, 54, 61 and 70) by performing a multiplex PCR with CFX96 thermal cycler (Bio-Rad, Hercules, CA, USA). Data analysis and interpretation were automated with Seegene viewer software, according to the manufacturer’s instructions.

### 2.6. MicroRNA Array and Bioinformatic Analyses

Agilent Platform was used for the expression profiling of microRNA along with microarray protocol v.3.1.1 (Agilent Technologies, Santa Clara, CA, USA). Dephosphorylation and labelling with Cyanine 3-pCp was performed of 50 ng total RNA (containing miRNA and spike-in controls). Afterwards, the Cy3-labeled RNA was purified using a Micro Bio-Spin P-6 Gel Column from Bio-Rad Laboratories, Inc. in Hercules, California, and hybridized at 55 °C for 20 h on an 8 x 60K Human microRNA microarray slide from Agilent Technologies, which contains 2549 human microRNAs. After being cleaned, the slides were scanned using an Agilent Technologies G2565CA scanner, and the pictures were then obtained using Feature Extraction software version 10 (Agilent Technologies). The Gene Expression Omnibus (GEO) received a request for a GEO number and microarray raw data.

Hierarchical Clustering Comparison between data were evaluated by fold changes. For all microRNA end lies, a volcano plot and t-statistics analysis were performed, considering a *p*-value ≤ 0.05 to be significant, and those miRNAs had a deregulation of more than 1.5-folds (logFC ≥ 0.6).

The analysis of the differentially expressed microRNAs was performed using R [21] and RStudio [22] with limma [23] and gplots [24] packages, *p*-values were adjusted for false discovery rates using the Benjamini and Hochberg method. miRNet [25], miRTarBase v8.0 [26] and MITHrIL [27] was used to predict target genes, gene ontology was extracted using Panther DB [28] and the KEGG pathway was calculated with DIANA-miRPath v3.0 [29] and SPECifIC [30].

Comparisons between data were evaluated by the fold changes. For statistical comparison the miRNA expression was scaled using log2 and tested using Student’s t-test. The significance of the gene ontology was calculated by DIANA-miRPath v3.0, which extracts *p*-value using miRNA sampling simulations.

### 2.7. Evaluation of qPCR-Based miRNA Expression Analysis

Real-time qPCR was used to validate the microarray findings for miR-4485 5p, miR-4485 3p, and miR-4497. Primer sequences were located using http://www.ncbi.nlm.nih.gov/tools/primer-blast/ (accessed on 2 January 2021) database to identify them. The Superscript II Reverse Transcription Kit was used to create the cDNAs (Invitrogen, Carlsbard, CA, USA). Amplicons whose identities were verified by melting curve analysis were identified using SYBR GREEN fluorescent tracers. In a Rotor-Gene 3000, ICR was carried out (Corbett Research, Mortlake, Australia). Each reaction was carried out in a 50 uL reaction volume with 10x PC buffer, 50 mM MgCl2, dNTM mix, primerA and primerS at 10 uM and 10 uM, Platinum^®^ Taq DNA polymerase from Invitrogen, DNA diluted to 1:10, and SYBR GREEN@ from Invitrogen. The thermal denaturation profile involved 45 cycles of PCR at 94 °C for 45 s, gene-specific temperature annealing for 30 s, and the elongation at 72 °C for 30 s. Hot-start enzyme activation took place at 95 °C for 2 min. The housekeeping gene RNU6 was used to standardize gene expression. Every sample was examined three times, and the outcomes were presented as the relative expression intensities, as determined by the initial positive amplification cycle. The fold change was computed using the delta–delta Ct technique. Raw fluorescence data (Rn values) were exported for further analysis along with the Cq values generated automatically by the SDS program (threshold value = 0.2, baseline setting: cycles 3–15). Each sample was tested in triplicate and the results were expressed as relative gene expression intensities, as obtained from the first positive amplification cycle (Ct). qPCR data were expressed as means ± SD of 3 replicates, and differences between groups were evaluated by Student’s *t* test for unpaired data.

## 3. Results

### 3.1. MicroRNA Expression Profileas Evaluted by Microarray and Bioinformatic Analyses

The overall trend of microRNA expression in human cervical intraepithelial samples in cervix bearing HPV infection as related to the severity of the intraepithelial neoplasia (CIN 1–3) was determined by heatmap analysis (Figure 1).

The expression of the top 30 microRNAs with the lowest p-value per single comparison class (for example CIN 3 vs. CIN 1), also including the three statistically significant different microRNAs, is reported in Figure 2.

The classification of microRNAs in cervical intraepithelial neoplasia samples highlights the expression of fifteen microRNAs with a logFC ≥ 0.6 for each single sample (including the three statistically significant microRNAs) Figure 3.

No statistically significant differences were found in CIN 3–CIN 2 groups (Figure 3A). Four miRNAs (miR-4485-5p, miR-4485-3p, miR-4497, miR-4507) are commonly down-regulated among CIN 3/CIN 2 group compared to CIN 1 group (Figure 3A).

Our results demonstrates that 15 out of the 2549 tested microRNAs were altered in CIN 3–2 tissues biopsies as compared to CIN 1 (Table 1). Of those miRNAs, miR-4485-5p, miR4485-3p and miR-4497 were down-regulated more than 1.5-fold and above the statistical significance threshold of *p* < 0.05 (Figure 4).

The qPCR validation of miRNA microarray data is reported in Figure 5 in which the amplification curves for each sample, either CIN 3 (light blue) or CIN 1 (purple). The relative miRNAs expression intensities were: (a) miR-4485 3p, 2.7 + 1.4 in CIN 3 and 5.6 + 2.6 in CIN 1 (*p* < 0.05); (b) miR-4485-5p 1.9 + 1.2 in CIN 3 and 4.6 + 1.3 in CIN 1 (*p* < 0.05); and (c) miR-4497 1.2 + 1.4 in CIN 3 and 3.4 + 1.9 in CIN 1 (*p* < 0.05). This pattern resembles the microarray-detected 1.5-fold down-regulation in CIN 3 as compared to CIN 1 (see Figure 2).

### 3.2. HPV Genotyping, Viral Load, and Co-Infections as Evaluated by Multiplex PCR

One of the high-risk HPV genotypes was detected in all samples. Histological analysis identified five patients with CIN 1 lesions, four with CIN 2 and seven with CIN 3. HPV16 has been found as a single or multiple infection in both low-risk (CIN 1) and high-risk (CIN 2–CIN 3) lesions. HPV16 is the prevalent genotype and was detected in the lesions of 7 out of 22 women (Table 2).

For ki-67, only nuclear positivity was taken into consideration. Immunoreactivity was assessed as occupying either the lower one third (1/3), lower two thirds (2/3) or all three thirds (3/3) of the epithelium.

The table shows the genotypes detected by molecular analysis. They are divided into high and low risk type. The “+” indicate the amount of viral load detected (see also Figure 5), which are between 1 and 3. The co-infection is shown by the presence of multiple genotypes in the same patient, which can be high and/or low risk type. The presence of co-infection and a high viral load identify a lesion with a high risk of neoplastic progression.

The data reported in (Table 1) show nine out of twenty-two cases with co-infections and the majority infected by high-risk HPV genotypes. Such patients have a greater risk of neoplastic progression and should be monitored more carefully over time than other cases.

**Figure 6 jpm-13-00531-f006:**
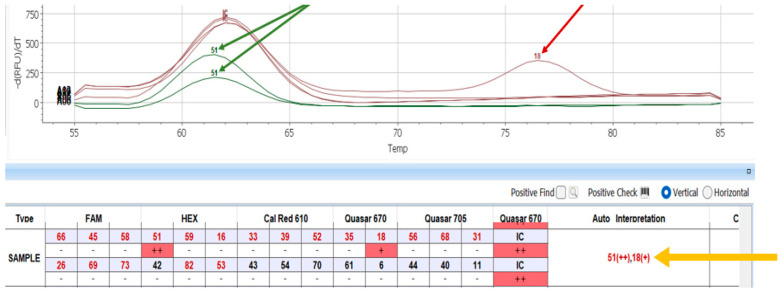
Analysis of genotype, viral load and coinfections of HPV samples as detected by melting curve analysis. In box “Auto Interpretation”, indicated by yellow arrow, the analysis data for each HPV sample are reported. The selected sample has co-infection (HPV51 and HPV 18), each with a different viral load values indicated with + symbols. Viral load evaluation is semi-quantitative and is based on the number of melting curve florescence detections: the greater the number of detections, the greater the viral load value, and it is reported as ++ or + or − (if not detected).

The detection of viral load is semi-quantitative and is reported in the graph (upper part). In particular, the thermal profile provides for three fluorescence measurements at the three melting temperatures, i.e., at steps 8, 14 and 20. If the fluorescence related to that genotype is detected all three times, the sample have a viral load equal to +++. Similarly, if it is detected only twice, it will be ++. If it is detected only once, it will be +. The selected sample has an HPV 51 and HPV 18 co-infection (indicated by yellow arrow), each with a different viral load. HPV 51 has been detected twice (green line) and, therefore, has viral load indicated with ++; while HPV 18 has been detected only once (red line) and has viral load indicated with + (Figure 6). 

### 3.3. Immunohistochemistry

Out of the twenty-two cervical intraepithelial neoplasia, ten cases were proven to be HPV positive (73%) with p16 immunohistochemical staining (Figure 7). Genotyping found seven cases of HPV 16 positivity with a combined positivity of HPV 16 and ki67 high score, indicating a high cell proliferation rate (Figure 7). 

The difference in expression levels of miR-6085, miR-6749-5p, miR-6875-5p and miR-7107-5p in positive or negative expression for p16 and CIN levels is reported in Figure 8. We used eighteen samples in the categories p16+ (seven CIN 2 and nine CIN 3) and two samples in the category p16- (CIN 1). The analysis shows an increased expression of miR-6085, miR-6749-5p and miR-7107-5p in p16-positive CIN 2 and CIN 3 compared to p16-negative CIN 1. In contrast, miR-6875-5p is highly expressed in p16-negative CIN 1 compared to p16-positive CIN 2 and CIN 3.

The terms in the Gene Ontology for the down-regulated microRNA were primarily associated with biological control and cellular processes, including cell adhesion molecules (Table 3). Pathway analysis revealed that the down-regulated genes were linked to numerous important biological processes, including transcriptional factors, ECM-receptor interactions and cancer-related pathways.

The significant pathways of the differentially expressed genes according to the KEGG database were found using pathway analysis.

KEGG analysis for the statistically significant and down-regulated microRNAs between CIN 2–3 and CIN 1. (Figure 9).

## 4. Discussion

The role of microRNAs has been reported in cervical cancer. However, it is also well established that cervical cancer is caused due to persistence infection with high risk-HPV [14]. microRNA down-regulation is the contributing factor involved in carcinogenesis and disrupts the function of p53 gene which regulates the post-transcriptional maturation of microRNAs [13]. The E6 and E7 of HPV, can modify the expression of molecules involved in the regulation of cell protein expression, such as microRNAs [31]. All these events lead to genetic instability, which increases the risk of random mutations and cell damage.

The most frequent genotype is HPV 16, mainly associated with high-risk lesions (CIN 2/CIN 3), confirming what is reported in numerous literature studies [9,10].

In the pool analyzed, only one case presenting HPV 16 infection is a low-risk lesion (CIN 1), but, due to the infecting genotype, it should be closely monitored over time for the high risk of neoplastic progression.

The infection process begins with the HPV penetration phase at the level of the host’s multilayered squamous epithelia. Initially, the viral genome is maintained in episomal form with respect to the host cell genome. The crucial event for neoplastic progression is represented by the integration of the viral genome into that of the host cell. Integration causes a strong increase in the expression of the oncogenic proteins E6 and E7 [6,32,33].

If the infection is persistent, high-risk HPV E6 and E7 can promote DNA damage through interaction with p53 and pRb-E2F, resulting in cell cycle alteration, dysplastic and then neoplastic transformation [7,8]. Furthermore, the inactivation of pRb by E7 results in the overexpression of the protein p16 (inhibitor of cyclin-dependent kinases) in dysplastic cervical cells, which is considered a useful biomarker of integration of the viral genome easily detectable by immunohistochemistry [9]. The accuracy of the histological examination can be improved through the analysis of p16 together with Ki-67. The analysis of the immunohistochemical expression of p16/Ki-67 highlights the presence of dysplastic cells transformed by HPV and the integration of the viral genome [10]. It is not yet clear which factors determine the malignant fate of a high-risk HPV infection, but in the last decade, attention has been focusing on the epigenetic alterations that underlie the progression to cancer. Many studies indicate that microRNA deregulation contributes to cervical cancer tumorigenesis [34]. The importance of microRNAs in cervical tumors is linked to the fact that the microRNA loci are associated with fragile sites, known as the insertion sites of the HPV virus in cervical tumors. Furthermore, the genes encoded by the virus can influence the expression of microRNAs in cervical cells. In general, microRNAs can regulate both tumor suppressor and oncogenes genes and the altered expression of microRNAs represents an early event in the induction of carcinogenesis by HPV infection [7]. Furthermore, regarding the gene expression profile of cervical carcinoma, an amplification of chromosome 5p was found and the up-regulated genes in this area are represented by the RNAsiIII Drosha complex responsible for the processing of microRNAs [35]. Although oncogenic genotypes are associated with cervical cancer, HPV infection alone is not sufficient to induce malignant and cervical cancer transformation, which implies also considering multilevel molecular factors, such as viral integration status, genotyping, viral load, and microRNA expression profiles.

We identified three microRNAs, including miR-4497, miR-4485-3p, and miR-4485-5p, that are differentially expressed in HPV in the high-risk CIN 3 samples compared to HPV subject with worst lesions CIN 1.

Lastly, microRNAs from the intracellular pool, that is, hsa-miR-664a-3p, hsa-miR-664a-5p, hsa-miR-664b-3p, hsa-miR-4485-3p, hsa-miR-10527-5p, and hsa-miR-12136, and that from the exosomal pool, that is, hsa-miR-7704, were up-regulated in vascular smooth muscle cells during replicative senescence (*n* = 3, FDR < 0.05) [17,36].

In the ovarian cancer cell line, the up-regulation of miR-221-3p, miR-222-3p, and miR-4485 and the decreased expression of miR-551b-3p, miR-551b-5p, and miR-218-5p were analyzed [37]. Several microRNAs derived from the mitochondrial genome have been observed [38], suggesting that this lncRNA could constitute a precursor for miR-4485-3p [39] The risk of developing cancer and precancer varies greatly among HPV genotypes, with HPV16 and HPV18 being the most carcinogenic and responsible for 70% of cervical squamous cell carcinomas [40].

Thus, *KLF12* may play a major role in the underlying mechanisms that lead to high-risk HPV infection and in cervical carcinogenesis process [41]. *KLF12* expression was significantly down-regulated in patients with ovarian cancer, endometrial cancer, and cervical cancer [42]. The decreased expression of *KLF12* was observed in the nucleus of both cervical squamous cell carcinoma tissue and adenocarcinoma tissue. The expression of *KLF12* was decreased in ovarian cancer and endometrial cancer, suggesting its role as a biomarker for gynecological tumor monitoring [43]. This study demonstrates, with an underlying lesson, that in most high-risk HPV positive cases, the genotype is detected by the HPV screening.

In KGN cells, MiR-1224-5p functioned by directly targeting FOXO1 and negatively regulating FOXO1 expression [44,45]. In esophageal squamous cell carcinoma, miR-324-5p down-regulation is a prevalent change that stimulates cell division, migration, invasion, and tumor growth via triggering the EGFR-EFNA1/EPHA2-VEGFA signaling pathway by suppressing TNS4 expression.

The herein presented study has some limits. Given the small sample size and the large pool of mRNAs detected in microarray, the statistics can be highly variable, even though algorithms for false discovery rate corrections have been applied. Although there were few samples analyzed, the data obtained from microRNA analysis involved 2700 microRNAs for each subject analyzed, and this expression panel has never been explored before and can help to insert new predictive markers of the state of severity of the HPV cancer process. We considered the p16 indicator of viral integration, used in clinical practice, because its expression is related to HPV expression. The positive rate of P16 level in cervical squamous epithelial was extremely high in CIN, especially in high-grade CIN [46]. The biological function of identified miRNAs, as related to HPV infection and/or cervical lesions progression. is not easily identifiable, resulting in the fact that each miRNA is regulates a variety of biological functions. However, miRNA analysis is a further endpoint providing information that could be integrated with those provided by HPV genotypes, viral load, and IHC, thus increasing their predictivity.

The use of miRNA has certain pros or cons regarding its use compared to HPV genotyping by PCR and immunohistochemistry (IHC) analyses. HPV genotyping by PCR demonstrate the viral presence in the tissue, and of course, this is the major risk factor for cancer progression. However, the presence of HPV alone does not indicate those who are at a high risk for rapid cancer progression. IHC is a phenotypic analysis that identifies tissue alterations in the early stages of the carcinogenesis process [47]. The interaction between HPV 7d host tissue before and during the initiation of the histological lesions results in an early molecular event called a miRNA change. Indeed, the primary epigenetic driver of the carcinogenesis process are miRNA changes.

miRNA is considerably stable in collected specimens, since they are short (22–25 nucleotides) and hardly degraded by RNAases at variance with long messenger RNAs.

A limit of miRNA analysis is that miRNA, at variance with messenger RNA, are not univocally related to a single biological function but play a role in multiple biological function. Accordingly, miRNA cannot be used as a unique molecular end point to predict the risk of developing cancer. Because of this situation, a multiple miRNA analysis (miRNA signature analysis) may be used, or miRNA can be used in combination with other analyses such as PCR and IHC. Indeed, the use of multiple biomarkers is a reasonable strategy to increase the predictivity of the performed analysis in identifying at individual level the risk of CIN progression. Larger studies dealing with the efficacy of combined new predictive biomarkers for CC screening and their applicability for personalized prevention in women with HPV lesions are required. It will be necessary to show accuracy to support evidence-based recommendations on CC secondary prevention strategies for the female population. Predictive combined biomarkers could be useful before invasive diagnostic interventions, allowing the treatment only of lesions with a real risk of neoplastic progression.

In an era characterized by increasing economic pressure, health systems around the world are faced with challenges related to the need of guaranteeing to all citizens access to a high-quality healthcare. In the past, the decision to introduce new technologies was mainly dependent on their efficacy. Today it is not possible to ignore sustainability in the light of providing high-value healthcare [48].

## 5. Conclusions

It is well established that HPV16 and 18 infections provide have a dramatically increased risk for CIN 3 and cancer onset. However, the detection of viral nucleic acid does not provide any information dealing the biological result of the interaction between HOV and the human host. This piece of information is provided by miRNA analysis in cervical tissue. Obtained results indicate that the downregulation of miR-4485-3p, miR-4485-5p, and miR-4497 is a hallmark of advanced CIN lesions. This finding supports the combined use of HPV16/18 genotyping and microRNAs detection as a triage test for HPV positive women to identify subject at high risk for cancer progression. The characterization of the CIN by multiple methods (HPV genotyping, miRNA, IHC) is a new tool to identify subject at high risk for cancer evolution in advance. This would lead to the targeted treatment of those patients with a CIN at high risk of progression to invasive forms, thus personalizing both therapeutic and follow-up protocols.

## Figures and Tables

**Figure 1 jpm-13-00531-f001:**
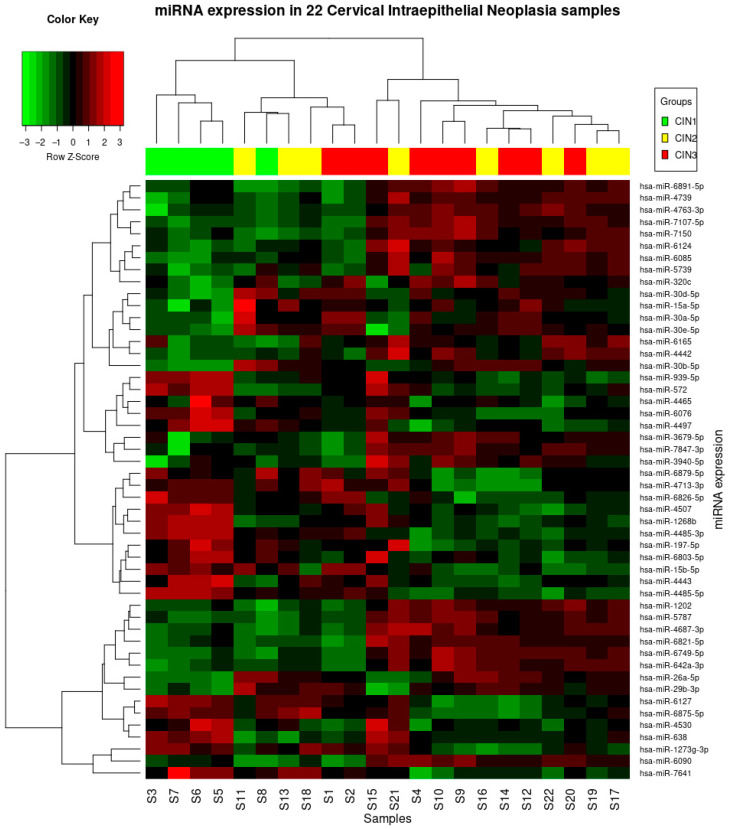
Heatmap comparing all analyzed samples (S1–S22) and showing a classification based on the expression values of the top 50 of all 2549 microRNAs analyzed. At the top, under the green-yellow-red bar, the samples are grouped together according to their CIN 1–3 level.

**Figure 2 jpm-13-00531-f002:**
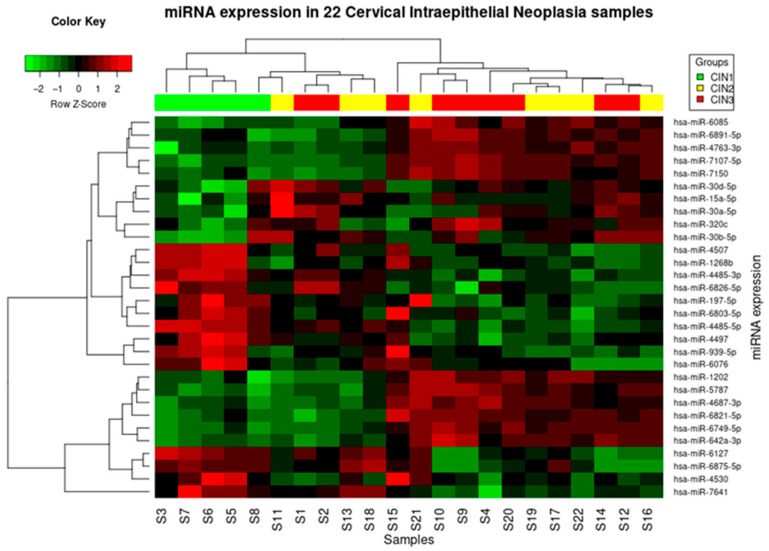
Heatmap for the top 30 deregulated miRNA. miR-4485-5p, miR-4485-3p and miR-4497-5p are the most statistically significant down-regulated in patients with grade CIN 3–CIN 2 compared to those with CIN 1. The *p*-values for the dysregulated miRs are has-miR-4485-5p, *p* = 0.002; hsa-miR-4485-3p, *p* = 0.002; hsa-miR-4497, *p* = 0.013. These microRNAs are downregulated with a logFC > 1.5.

**Figure 3 jpm-13-00531-f003:**
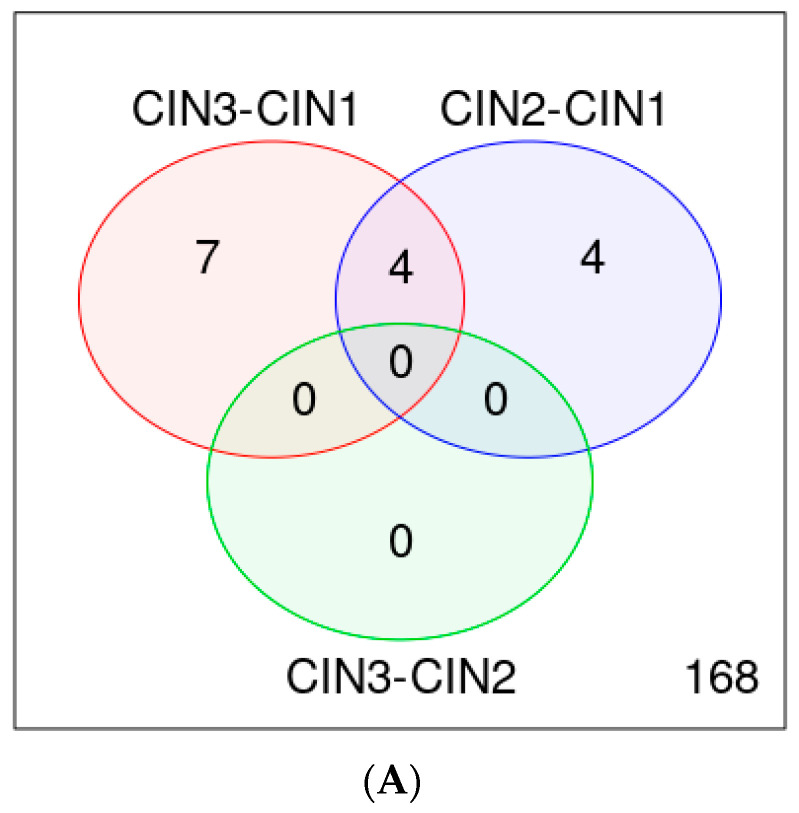
(**A**) Venn diagram analysis illustrates the relationships of statistically significant deregulation of microRNAs in the various CIN sets; the red circle shows a total of eleven significantly deregulated microRNAs between CIN 3 and CIN 1, of which four (intersection of red and blue circle) shared to those deregulated between CIN 2 and CIN 1 to which four microRNA exclusively belong (blue circle without intersection); however, no significant deregulation was found between CIN 3 and CIN 2 group (green circle). There are 15 significantly altered microRNAs. (**B**) Mean difference plot of deregulated microRNAs between CIN 3 versus CIN 1 seven blue and four red highlighted points correspond to the significantly altered microRNAs.

**Figure 4 jpm-13-00531-f004:**
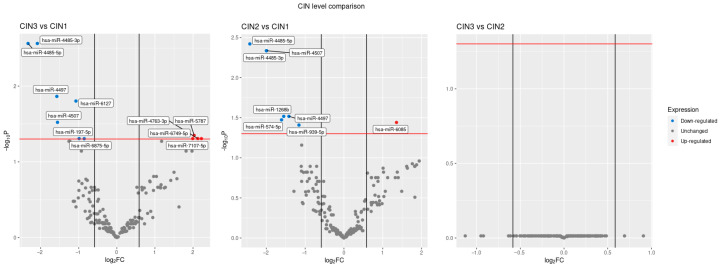
Volcano plot analysis underlines the expression of microRNA from left to right the volcano plots of deregulated miRNA in group differences: CIN 3 vs. CIN 1, CIN 2 vs. CIN 1 and CIN 3 vs. CIN 2. Vertical black lines represent a |log2(FC)| > 0.6 and the red horizontal line shows an adjusted *p*-value < 0.05. Blue and red dots are miRNAs down- and up-regulated, respectively.

**Figure 5 jpm-13-00531-f005:**
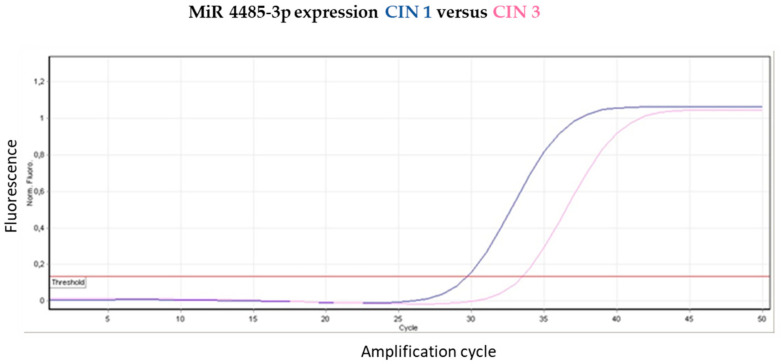
qPCR analysis of miRNAs. The panel’s report the amplification curves of the HPV samples tested, either CIN 1 (blue) or CIN 3 (pink) (upper panel) or CIN 1 (purple) or CIN 3 (light blue) (middle panel), relatively to the miRNAs miR-4485 5p or miR-4485 3p, and either CIN 1 (green) or CIN 3 (blue) (lower panel), relatively to miR-4497.

**Figure 7 jpm-13-00531-f007:**
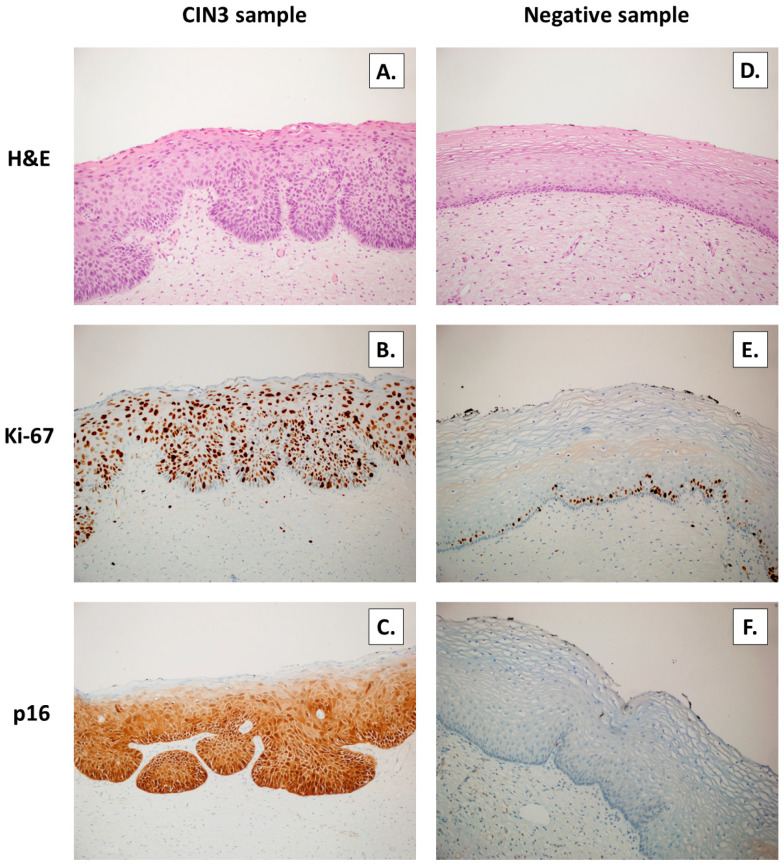
Cervical intraepithelial neoplasia CIN 3 lesions (**A**) and negative control, normal cervical sample (**D**) detected by hematoxylin–eosin (H&E) staining (×200); characterized by three-thirds extended nuclear expression of Ki-67 (**B**) with ki-67 expression limited to the basal layer (**E**) detected by immunohistochemistry by anti-Ki67 (×200); strong, diffuse, nuclear and cytoplasmic expression of p16 (**C**); and no stain for p16 (**F**) was detected by immunohistochemistry by anti-p16 (×200).

**Figure 8 jpm-13-00531-f008:**
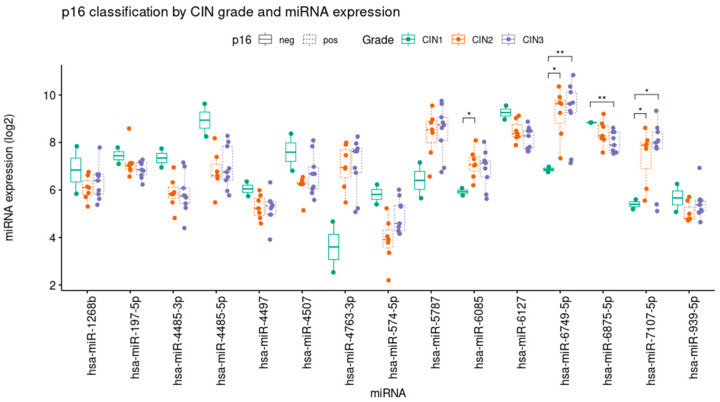
Expression levels of the 15 deregulated miRNA classified for CIN level and p16. miR-6085, miR 6749-5p, miR 6875-5p and miR-7107-5p differences in positive p16 and negative p16 are statistically significant. Histograms depicts the quantitative data of means + SE of positive p16, *p* < 0.05 vs. p16 negative samples. * *p* < 0.5, ** *p* < 0.05.

**Figure 9 jpm-13-00531-f009:**
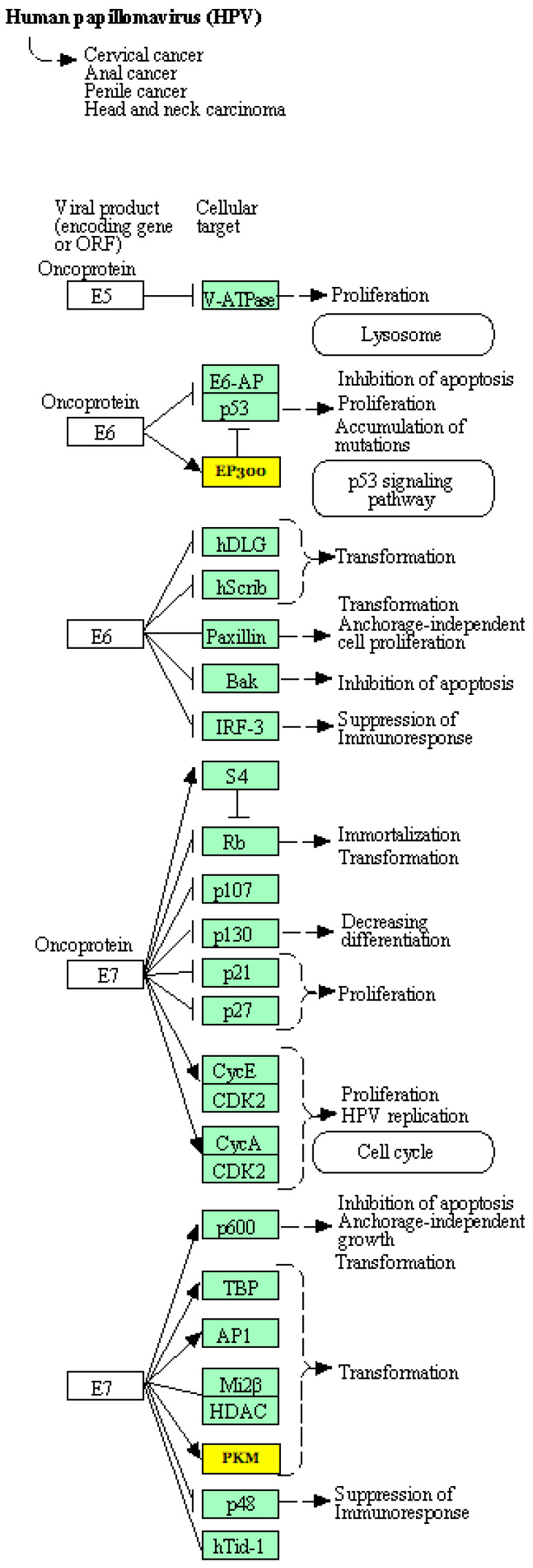
KEGG pathway analysis of genes targeted by the HPV carcinogenesis. According to their P-values, the top-ranking canonical KEGG pathways discovered are listed in list 27. Significant enrichment of route 27d was observed (*p* =0.05). The EP300 and PKM genes are both targets of hsa-miR-574-5p.

**Table 1 jpm-13-00531-t001:** Summary of FC, logFC and p-value for statistically significant miRNA.

CIN 3–CIN 1	FC	logFC	*p*	CIN 2–CIN 1	FC	logFC	*p*
miR-4485-5p	−5.03	−2.330	0.003	miR-4485-5p	−5.37	−2.425	0.004
miR-4485-3p	−4.25	−2.086	0.003	miR-4485-3p	−3.99	−1.999	0.005
miR-4497	−2.98	−1.574	0.014	miR-4507	−3.98	−1.994	0.005
miR-6127	−2.10	−1.073	0.016	miR-4497	−2.67	−1.418	0.030
miR-4507	−2.94	−1.556	0.030	miR-1268b	−2.93	−1.551	0.030
miR-5787	4.08	2.030	0.045	miR-574-5p	−3.04	−1.608	0.034
miR-7107-5p	4.66	2.220	0.049	miR-6085	2.56	1.355	0.036
miR-6749-5p	3.97	1.992	0.049	miR-939-5p	−2.23	−1.159	0.039
miR-4763-3p	4.36	2.125	0.049				
miR-197-5p	−1.99	−0.990	0.049				
miR-6875-5p	−1.81	−0.856	0.049				

**Table 2 jpm-13-00531-t002:** Clinico-pathological characteristics of cervical specimens and analysis of their genomic DNA for HPV infection.

Tissue Type	Characteristics	Molecular Analysis		p16	ki-67
		High Risk	Low Risk		
Cone	CIN 3	69+		pos	3/3
Biopsy	CIN 3	16++		pos	3/3
Biopsy	CIN 1	69+		neg	1/3
Cone	CIN 3	51+ 53++	61+	pos	3/3
Biopsy	CIN 1	16++		--	--
Pap test	CIN 1	51+ 53++		--	--
Biopsy	CIN 1	18+ 51++		--	--
Biopsy	CIN 1	52+ 56+ 73+	42++ 6+++ 40++	neg	1/3
Cone	CIN 3	16+		pos	3/3
Cone	CIN 3	16++		pos	3/3
Cone	CIN 2	33++		pos	3/3
Cone	CIN 3	52++		pos	3/3
Biopsy	CIN 2	66+	54++ 40+++	neg	1/3
Cone	CIN 3	18++		pos	1/3
Biopsy	CIN 3	66++ 16++		pos	2/3
Biopsy	CIN 2	16++		pos	1/3
Cone Cone	CIN 2 CIN 2	- -	- -	Pos pos	3/3 3/3
Cone Cone Cone Cone	CIN 2 CIN 3 CIN 2 CIN 2	56+ 33+ 16++ -	- - - -	Pos Pos Pos pos	3/3 3/3 3/3 1/3

Evaluation of viral load values determined by semi-quantitative scale: + low, ++ medium, +++ high. Evaluation of p16 staining was performed as positive (“pos”) and negative (“neg”) reaction. For p16 it was evaluated as positive nuclear or cytoplasmic staining. Nuclear staining was the only aspect of Ki-67 scoring, and it was given a score of 0 (no staining), + (focal basal/parabasal staining), ++ (diffuse staining restricted to the bottom third) and + (diffuse staining of the whole epithelium).

**Table 3 jpm-13-00531-t003:** Gene Ontology.

GO Category	GO Name	GO Identifier	*p*-Value	Genes	miRNAs
CC	organelle	0043226	2.879 × 10^−11^	191	3
BP	cellular nitrogen compound metabolic process	0034641	1.189 × 10^−7^	102	3
MF	ion binding	0043167	8.693 × 10^−5^	115	3
BP	viral process	0016032	0.006 × 10^−1^	17	3
BP	symbiosis encompassing mutualism through parasitism	0044403	0.007 × 10^−1^	18	3
BP	biosynthetic process	0009058	0.007 × 10^−1^	80	3
BP	gene expression	0010467	0.004	17	3
BP	membrane organization	0061024	0.004	19	3
BP	catabolic process	0009056	0.011	43	3

## Data Availability

The datasets used and/or analyzed during the current study are available from the corresponding author on reasonable request.

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
