# Peer review of "Screening of Precancerous Lesions in Women with Human Papillomavirus (HPV) Infection by Molecular Typing and MicroRNA Analysis"

_jpm, 2023, doi:10.3390/jpm13030531_

Round 1

Reviewer 1 Report

This study is a meticulous translational research for applying microRNA detection to the diagnosis of CINs; however, there are several problems in the paper, which needs revisions.

1. I consider that Fig. 3 is insufficient, because signs of A and B are not shown in the figures. In addition,  it is difficult to understand what CIN3-CIN1, CIN2-CIN1, CIN3-2 means in Fig. 3A, even after I read the description.

2. In the middle of discussion, the following sentence suddenly apperes.  The uppercase and lowercase typeface are different from other sentences. You may be suspected of copy & paste from somewhere, so I reccomend to correct it.

“Human Papillomavirus Type 16 Expression, Molecular Characteristics, and Clinical Outcome are Closely Associated with Immune Signature Based Subtypes of Cervical Squamous Cell Carcinoma. “

3. KLF12 suddenly appears in the discussion part, but the explanation of KLF12 is insufficient.

4. The authors state about the significance of this study that their finding supports the combined use of HPV16/18 genotyping and microRNAs detection as triage test for HPV positive women to identify subject at high risk for cancer progression. On the other hand, in order for this method to be implemented as a clinical test, it is necessary to show accuracy such as the sensitivity and specification using the triage manifested, and I think that the discussion about cost-effectiveness and affordability as a new testing should be included in the Discussion.

Author Response

Dear Editors,

We would like to thank you for considering the manuscript entitled “Screening of Precancerous Lesions in Women with Human Papillomavirus (HPV) Infection by Molecular Typing and MicroRNA Analysis” by Varesano S., Pulliero A.  et al. and for sharing the Reviewers’ comments that certainly helped in improving the quality of the manuscript (jpm-2229597). We appreciated the Reviewers’ comments, and we revised the manuscript accordingly. Please find enclosed to the submission of the revised version of the manuscript the point-by point reply to the Reviewers’ comments. For clarity’s sake, changes in the revised MS are marked in yellow.

We hope that the revised version of our MS will be now suitable for publication in the JPM.

Accordingly, we prepared a revised version of the manuscript acknowledging Referees’ and Editor’s comments as below specified:

Reviewer 1:

COMMENT 1.

I consider that Fig. 3 is insufficient, because signs of A and B are not shown in the figures. In addition, it is difficult to understand what CIN3-CIN1, CIN2-CIN1, CIN3-2 means in Fig. 3A, even after I read the description.

ANSWER 1. We are grateful with the Reviewer for the suggestions. The Figure 3 has been completed and the Figure Legend description implemented as requested.

COMMENT 2. In the middle of discussion, the following sentence suddenly apperes. The uppercase and lowercase typeface are different from other sentences. You may be suspected of copy & paste from somewhere, so I reccomend to correct it.

ANSWER 2. We thank the Reviewer for the note. We have corrected it. The lowercase text was a reference, now we have corrected with the reference number.

COMMENT 3. KLF12 suddenly appears in the discussion part, but the explanation of KLF12 is insufficient.

ANSWER 3. We are grateful with the Reviewer for the suggestions. We added some references to support the role of KLF12.

COMMENT 4. The authors state about the significance of this study that their finding supports the combined use of HPV16/18 genotyping and microRNAs detection as triage test for HPV positive women to identify subject at high risk for cancer progression. On the other hand, in order for this method to be implemented as a clinical test, it is necessary to show accuracy such as the sensitivity and specification using the triage manifested, and I think that the discussion about cost-effectiveness and affordability as a new testing should be included in the Discussion.

ANSWER 4. We thank the Reviewer for the note. The requested part has been added at the end of the Discussion section.  

Reviewer 2 Report

The authors have studied the pattern of different miRNAs in nearly 20 cervical cancer patients with HPV infection. The data, graphs and analysis looks good, however the results need to be more systematic and scrutinized. 

1. The precancerous lesions (CIN1 to CIN3) should be elaborate more in the introduction or method section with specific criteria. The 22 cases should be classified into these 3 categories. What are the differences observed as per miRNA seq analysis in these groups? 2. Line 248: logFC>0.6 barely seems to be a threshold value to see significant gene expression differences. At least FC>2 should be considered. List all the significant genes with FC and P-value.  3. Fig 8: the p16 classification in positive and negative groups data should be analyzed more vigorously. What is the conclusion of this graph? 4. Fig 5: The qPCR melting curves are good to see the difference in miRNA expression, but graphs must be plotted with p-value to check differences among different groups.

Author Response

Reviewer 2:

COMMENT 1. The precancerous lesions (CIN1 to CIN3) should be elaborate more in the introduction or method section with specific criteria. The 22 cases should be classified into these 3 categories. What are the differences observed as per miRNA seq analysis in these groups?

ANSWER 1. In the Introduction, the classification of the 3 categories has been added. Furthermore, a description of CIN 1 to CIN 3 lesions is now reported in the newly added paragraph (15 lines). We find statistically significant differences in 15 miRNA between the CIN3-CIN1 and CIN2-CIN1 groups. No statistically significant differences were found in CIN3-CIN2 groups. Four miRNAs (miR-4485-5p, miR-4485-3p, miR-4497, miR-4507) are commonly down-regulated among CIN3/CIN2 group compared to CIN1 group. The result shows that CIN3/CIN2 are more similar in miRNA expression than those in CIN1 samples.

COMMENT 2. Line 248: logFC>0.6 barely seems to be a threshold value to see significant gene expression differences. At least FC>2 should be considered. List all the significant genes with FC and P-value.

ANSWER 2. Thanks for the observation, as commonly used in differential expression analysis, logFC is a base 2 logarithm of the fold change (log2(FC)). A cutoff of |FC|>2 is equivalent to a log2(|FC|)>1, where |x| denote the absolute value of x to make sure that the log value can be calculated. Since Tab.1 already shows all the logFC>0.6 and its p-value, we decide under your suggestion to describe it better by adding a column for FC, (see Table 1 in yellow in the text) .

COMMENT 3. Fig 8: the p16 classification in positive and negative groups data should be analyzed more vigorously. What is the conclusion of this graph?

ANSWER 3. We are grateful with the Reviewer for the suggestions. We have performed a more rigorous analysis of p16 classification and updated the relative paragraph in the manuscript. Here we have compared miRNA expression based on p16 status and CIN level. We noted an increased expression of miR-6085, miR-6749-5p and miR-7107-5p in p16-positive CIN2 and CIN3 compared to p16-negative CIN1. In contrast, miR-6875-5p was highly expressed in p16-negative CIN1 compared to p16-positive CIN2 and CIN3. However, as can be seen in the lower part of the figure we have only 2 patients with CIN1 grade and p16-negative. The Figure 8 has been changed, it is now clearer.

COMMENT 4. Fig 5: The qPCR melting curves are good to see the difference in miRNA expression, but graphs must be plotted with p-value to check differences among different groups.

ANSWER 4. We thank the Reviewer for the note. The p- value and the relative miRNAs expression intensities have been added in the Results section.  

Round 2

Reviewer 2 Report

.